# Astigmatic-Invariant Structured Singular Beams

**Alexander Volyar [1],\*, Eugeny Abramochkin [2] , Yana Akimova [1] and Mikhail Bretsko [1]**

1   V.I. Vernadsky Crimean Federal University, Vernadsky Prospect 4, 295007 Simferopol, Russia
2   Lebedev Physical Institute, 443011 Samara, Russia
*   Correspondence: volyar.singular.optics@gmail.com

**Abstract:** We investigate the transformation of structured Laguerre–Gaussian (sLG) beams after passing through a cylindrical lens. The resulting beam, ab astigmatic structured Laguerre–Gaussian (asLG) beam, depends on quantum numbers $(n, \ell)$ and three parameters. Two of them are control parameters of the initial sLG beam, the amplitude $\epsilon$ and phase $\theta$. The third one is the ratio of the Rayleigh length $z_0$ and the focal length $f$ of the cylindrical lens. It was theoretically revealed and experimentally confirmed that the asLG beam keeps the intensity shape of the initial sLG beam when the parameters satisfy simple conditions: $\epsilon$ is unity and the tangent of the phase parameter $\theta/2$ is equal to the above ratio. We also found sharp bursts and dips of the orbital angular momentum (OAM) in the asLG beams in the vicinity of the point where the OAM turns to zero. The heights and depths of these bursts and dips significantly exceed the OAM maximum and minimum values of the initial sLG beam and are controlled by the radial number $n$.

**Keywords:** vortex beams; structured light; orbital angular momentum; astigmatic-invariant beams





## 1. Introduction

The unique ability of structured singular beams [1,2] to contain multiple degrees of freedom is their undoubted advantage over single vortex and vortex-free Laguerre–Gaussian (LG) beams, Hermite–Gaussian (HG) beams, Ince–Gaussian (IG) beams and other beam types [3,4]. The ability to control degrees of freedom reveals the unrestricted possibilities of their applications in optical communication [5], optical tweezers [6], quantum entanglement [7] and others. As a rule, amplitudes and phases of standard singular modes in structured beams are strongly matched, which makes them structurally stable to propagation in free space [8–11], and energy flow control contributes to their structural stability [2,12], external perturbations and transitions to new structurally stable states [13].

We start with a standard LG mode representation in the HG mode basis [14]:

$$\mathrm{LG}_{n,\pm\ell}(\mathbf{r}) = \frac{(-1)^n}{2^{2n+3\ell/2}n!} \sum_{k=0}^{2n+\ell} (\pm 2\mathrm{i})^k P_k^{(n+\ell-k,n-k)}(0)\mathrm{HG}_{2n+\ell-k,k}(\mathbf{r}). \tag{1}$$

Here $\mathbf{r} = (x, y)$ is a 2D vector; $P_k^{(n,m)}(\cdot)$ is a Jacobi polynomial, $n$ and $\pm\ell$ are radial and azimuthal numbers of the LG beam; $\mathrm{LG}_{n,\pm\ell}(\mathbf{r}) = \exp(-|\mathbf{r}|^2)(x \pm \mathrm{i}y)^\ell L_n^\ell(2|\mathbf{r}|^2)$ and $\mathrm{HG}_{n,m}(\mathbf{r}) = \exp(-|\mathbf{r}|^2)H_n(\sqrt{2}x)H_m(\sqrt{2}y)$ are complex amplitudes of LG and HG modes, respectively. For simplicity, we will consider only the case $+\ell$ below.

Each HG mode in the wave composition (1) can be treated as a separate degree of freedom. The simplest way to transform the LG beam into a new one without losing its structural stability is to insert an extra factor into the sum as an excitation of each HG mode. We employ a two-parameter function, $\epsilon_k(\epsilon, \theta) = 1 + \epsilon e^{\mathrm{i}k\theta}$, so that Equation (1) turns into the expression which we consider as a definition of structured LG (sLG) beams:

$$\mathrm{sLG}_{n,\ell}(\mathbf{r}, \epsilon, \theta) = \frac{(-1)^n}{2^{2n+3\ell/2}n!} \sum_{k=0}^{2n+\ell} (2\mathrm{i})^k P_k^{(n+\ell-k,n-k)}(0)\epsilon_k(\epsilon, \theta)\mathrm{HG}_{2n+\ell-k,k}(\mathbf{r}), \tag{2}$$

where $\epsilon$ is an amplitude parameter and the parameter $\theta$ represents the initial phase. Such a transformation is easy to implement in practice using a spatial light modulator (SLM) [15]. This new family of paraxial beams has a number of unexpected properties. First of all, a direct but cumbersome algebra [16] turns it into a sum of two standard hybrid Hermite–Laguerre–Gaussian (HLG) beams [8] with different control parameters:

$$\mathrm{sLG}_{n,\ell}(\mathbf{r},\epsilon,\theta) = \frac{(-1)^n}{2^{n+\ell}n!}\{\mathrm{HLG}_{n+\ell,n}(\mathbf{r}|\pi/4) + \epsilon(-\mathrm{i})^{n+\ell}\mathrm{e}^{\mathrm{i}(2n+\ell)\Theta}\mathrm{HLG}_{n,n+\ell}(\mathbf{R}_{-\pi/4}\mathbf{r}|\Theta)\}, \qquad (3)$$

where $\Theta = \theta/2 - \pi/4$ is a control parameter of the HLG beam and $\mathbf{R}_\alpha = \begin{pmatrix} \cos\alpha & -\sin\alpha \\ \sin\alpha & \cos\alpha \end{pmatrix}$ is a rotation matrix at the angle $\alpha$. It is known that the LG-mode family is a particular family of HLG modes, as it written in Equation (A11). Thus, the structured LG beam, as a result of data excitation in each HG mode, can be treated as a sum of the LG mode and the HLG mode rotated by an angle $\alpha = -\pi/4$ and the control parameter $\Theta = \theta/2 - \pi/4$ with the same initial numbers $n$ and $\ell$. It is important to note that the variation of the control parameter causes fast oscillations in the orbital angular momentum (OAM), which corresponds to fast transitions of the sLG beam into various structurally stable states. Nevertheless, despite the fast oscillations of the OAM, the structured beam has a number of invariants in the form of conservation of the modulus of the total topological charge (TC). However, in the presented article, we raise the problem of the invariants of the sLG beam subjected to astigmatic transforms, for example, when passing through cylindrical lenses.

Generally speaking, the problem of astigmatic transformations of standard singular beams has already been considered by various authors (see, e.g., [2,14,17,18] and references therein). It was found that an astigmatic system of cylindrical lenses is capable of converting HG and LG beam states [19], and also contributes to measuring the TC of optical vortices [20,21]. Moreover, the rotation of the symmetry angle $\alpha$ of the cylindrical lens makes it possible to transform the HG beam into hybrid HLG beams, though simple astigmatism $\alpha = 0, \pi/2$ does not change the state of the HG beam [14,17,22]. However, this does not mean that the astigmatic transformation leaves unchanged a matched composition of HG modes in different states. Thus, the main aim of our study was to research a family of astigmatic-invariant sLG beams and unusual OAM transitions arising in them when the sLG beam is subjected to a simple astigmatism.

## 2. Astigmatic Transform of the sLG Beam and Its Invariants

Let a structured beam propagate through an astigmatic system with the astigmatic function in the form [14,17]

$$\psi(\boldsymbol{\rho},\phi) = (\xi^2 - \eta^2)\cos 2\phi + 2\xi\eta\sin 2\phi, \qquad (4)$$

where $\boldsymbol{\rho} = (\xi,\eta)$ is a 2D vector of coordinates in the astigmatic element (e.g., a cylindrical lens), and $\phi$ stands for an angular position of the cylindrical lens axes. Since the sLG beam is written as the sum of two hybrid HLG modes in Equation (3), it is convenient to use a general astigmatic transformation of an HLG beam in the form of the Fourier transform [17]. For a simple astigmatic function $\psi(\boldsymbol{\rho},0) = (\xi^2 - \eta^2)$ and the sLG beam, after some calculations (see Appendix A), we come to the complex amplitude of the astigmatic sLG (asLG) beam:

$$\frac{1}{2\pi}\int_{\mathbb{R}^2}\exp\{-\mathrm{i}\langle\mathbf{r},\boldsymbol{\rho}\rangle + \mathrm{i}b\psi(\boldsymbol{\rho},0)\}\mathrm{sLG}_{n,\ell}(\boldsymbol{\rho},\epsilon,\theta)\mathrm{d}^2\boldsymbol{\rho}$$
$$= \frac{(-\mathrm{i})^{2n+\ell}}{2\sqrt{1+b^2}}\exp\left\{-\frac{\mathrm{i}b\psi(\mathbf{r},0)}{4(1+b^2)} + \mathrm{i}(2n+\ell)\beta\right\}\mathrm{asLG}_{n,\ell}\left(\frac{\mathbf{r}}{2\sqrt{1+b^2}},\epsilon,\theta,\beta\right),$$
$$\mathrm{asLG}_{n,\ell}(\mathbf{r},\epsilon,\theta,\beta) = \mathrm{e}^{-\mathrm{i}(2n+\ell)\beta}\cdot\frac{(-1)^n}{2^{n+\ell}n!}\{\mathrm{e}^{\mathrm{i}\ell\pi/4}\mathrm{HLG}_{n+\ell,n}(\mathbf{R}_{-\pi/4}\mathbf{r}\,|\,\pi/4 - \beta)$$
$$+ \epsilon(-\mathrm{i})^{n+\ell}\mathrm{e}^{\mathrm{i}(2n+\ell)\Theta}\mathrm{HLG}_{n,n+\ell}(\mathbf{R}_{-\pi/4}\mathbf{r}\,|\,\Theta - \beta)\}, \qquad (5)$$

where $\beta = \arctan b$. We will call the parameter $\beta$ the astigmatic angle or simply the astigmatic parameter. It should not be confused with the angle $\phi$ in Equation (4), which is the control parameter of the hybrid HLG beam. It is known [17] that the angle $\phi$ sets the angle between the axis of the cylindrical lens and the axis $y$ in an experiment. For a simple astigmatism, the control parameter $\phi = 0$, but the angle $\beta$ is entered formally as $b = \tan \beta$ through the parameters of the astigmatic element $b$ (see the Experiment). Moreover, if the amplitude parameter is zero $\epsilon = 0$, then the astigmatic beam turns into a hybrid HLG beam (as in Equation (3)), but instead of the parameter, we use the astigmatic $\Theta$ parameter.

We intentionally introduce and separate the factors $e^{\pm i(2n+\ell)\beta}$ in the definition of the asLG beam, Equation (5), since then the beam description in terms of HG modes becomes similar to the sLG beam definition; see Equation (2):

$$\text{asLG}(\mathbf{r}, \epsilon, \theta, \beta) = \frac{(-1)^n}{2^{2n+3\ell/2}n!} \sum_{k=0}^{2n+\ell} (2\mathrm{i})^k P_k^{(n+\ell-k,n-k)}(0) e^{\mathrm{i}k(-2\beta)} (1 + \epsilon e^{\mathrm{i}k\theta}) \text{HG}_{2n+\ell-k,k}(\mathbf{r}). \tag{6}$$

It is easy to see that the astigmatic-free asLG beam (i.e., when $\beta = 0$) coincides with the initial sLG beam, $\text{asLG}(\mathbf{r}, \epsilon, \theta, 0) = \text{sLG}(\mathbf{r}, \epsilon, \theta)$. This can be deduced also from comparison of (5) and (3), by applying the relation $\text{HLG}_{n,m}(\mathbf{R}_\alpha \mathbf{r} \mid \pi/4) = e^{\mathrm{i}(n-m)\alpha} \text{HLG}_{n,m}(\mathbf{r} \mid \pi/4)$.

Although simple astigmatism does not change structure of a single HG beam (see Appendix A or reference [14,17]), the superposition of a large number of HG modes leads to rather radical transformations of the intensity pattern. This is due to appearance of the phase factors depending on the difference between HG mode indices; see Equation (A1). Since the factors are different for different HG modes, the intensity shape of asLG beams varies widely, as shown in Figure 1.

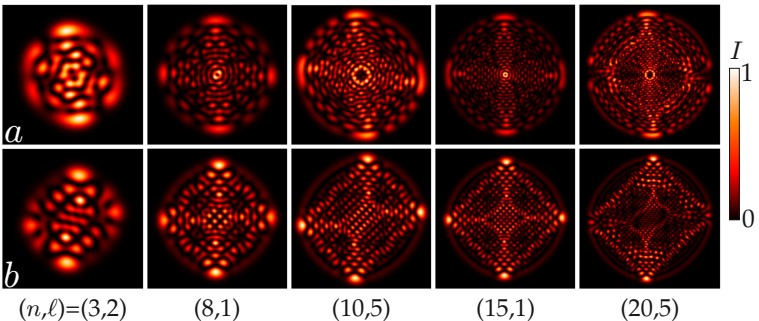

$(n,\ell)=(3,2)$　　　$(8,1)$　　　$(10,5)$　　　$(15,1)$　　　$(20,5)$

**Figure 1.** Computer simulation of the astigmatic transform of sLG beams (**a**) into asLG beams (**b**) with $\epsilon = 1$, $\theta = \pi/4$, and $\beta = \pi/3$ for some quantum number pairs $(n,\ell)$.

In order to obtain the conditions for astigmatic-invariant sLG beams, it is necessary to reduce the expansion (6) to the expansion (2). This may be done when $\epsilon = 1$ and

$$\theta = 2\arctan b = 2\beta. \tag{7}$$

Indeed, by substituting these values into Equation (6), we have

$$
\begin{aligned}
\text{asLG}(\mathbf{r}, 1, \theta, \theta/2) &= \frac{(-1)^n}{2^{2n+3\ell/2}n!} \sum_{k=0}^{2n+\ell} (2\mathrm{i})^k P_k^{(n+\ell-k,n-k)}(0) (e^{-\mathrm{i}k\theta} + 1) \text{HG}_{2n+\ell-k,k}(\mathbf{r}) \\
&= \left\{ \frac{(-1)^n}{2^{2n+3\ell/2}n!} \sum_{k=0}^{2n+\ell} (2\mathrm{i})^k P_k^{(n+\ell-k,n-k)}(0) (1 + e^{\mathrm{i}k\theta}) \text{HG}_{2n+\ell-k,k}(x, -y) \right\}^* \\
&= \left\{ \text{sLG}(x, -y, 1, \theta) \right\}^*.
\end{aligned}
\tag{8}
$$

This means that there will always be a pair of control parameters of the sLG beam, $\epsilon$ and $\theta$, with any radial $n$ and azimuthal $\ell$ numbers at which it turns into a beam of the same intensity pattern in a system with simple astigmatism $b$.

A comparison of typical intensity patterns of astigmatic-free sLG (a–c) and astigmatic-invariant asLG (d–f) structured beams in different states $(n, \ell)$ is presented in Figure 2, and the interference patterns demonstrate preservation of signs of the TC in the beam pairs. It is seen that the intensity patterns in pairs are the same except a mirror reflection, $y \to -y$. It is also important to note that the forked interference structure also remained the same, apart from small displacements of the fork centers. This indicates that the TC of the vortex modes in the asLG beam did not change their sign so that condition (7) holds for any pair of numbers $(n, \ell)$ of the asLG beams.

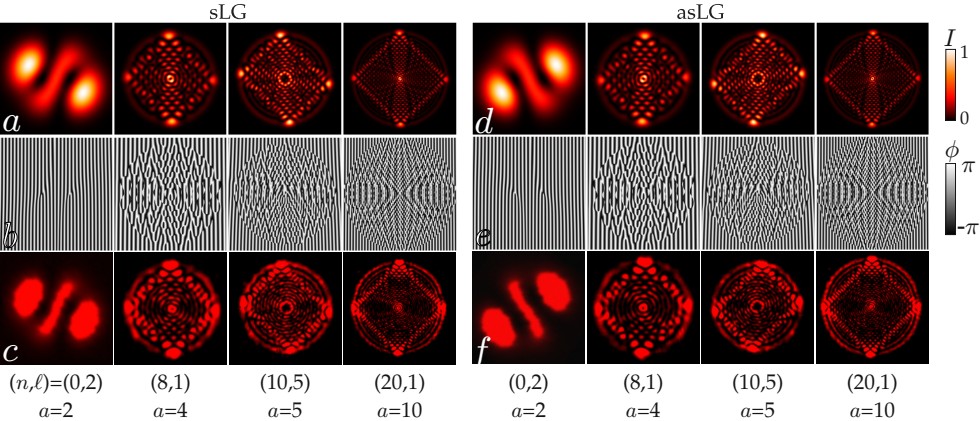

**Figure 2.** Theoretical (**a**,**d**) and experimental (**c**,**f**) intensity patterns of sLG and asLG beams, and a computer simulation of the interference patterns (**b**,**e**) for various states $(n, \ell)$ and the parameters $\epsilon = 1$, $\theta = \pi/2$, and $\beta = \pi/4$. Experimental intensity patterns are shown in a square $[-a, a] \times [-a, a]$, where the values of $a$ are pointed out in mm.

The OAM of any paraxial light field $F(\mathbf{r})$ may be found in various ways based on the standard definition, $\ell_z[F] = L/E$, where

$$L = \int_{\mathbb{R}^2} F^*(\mathbf{r})\{x\partial_y - y\partial_x\}F(\mathbf{r})\,d^2\mathbf{r}, \quad E = \int_{\mathbb{R}^2} |F(\mathbf{r})|^2\,d^2\mathbf{r}. \tag{9}$$

The ordinary way is to evaluate both integrals directly. Concerning asLG beams, this approach has definite benefits, since we can use Equation (5) to remove the common constant factor and the rotation of arguments. The final result is as follows:

$$\ell_z\big[\text{asLG}_{n,\ell}(\mathbf{r}, \epsilon, \theta, \beta)\big] = \ell \cdot \frac{\cos 2\beta + 2\epsilon \cos\big((2n+\ell)\frac{\theta}{2}\big) \cdot U + \epsilon^2 \cos(\theta - 2\beta)}{1 + 2\epsilon \cos\big((2n+\ell)\frac{\theta}{2}\big) \cdot V + \epsilon^2}, \tag{10}$$

where the factors $U$ and $V$ are expressed in terms of Jacobi polynomials:

$$U = \sin(\theta - 2\beta)\sin\tfrac{\theta}{2}\big(\cos\tfrac{\theta}{2}\big)^{\ell-1}\big\{P_n^{(1,\ell-1)}(\cos\theta) - P_{n-1}^{(1,\ell)}(\cos\theta)\big\} +$$
$$+ \cos(\theta - 2\beta)\big(\cos\tfrac{\theta}{2}\big)^{\ell}P_n^{(0,\ell)}(\cos\theta),$$
$$V = \big(\cos\tfrac{\theta}{2}\big)^{\ell}P_n^{(0,\ell)}(\cos\theta). \tag{11}$$

Details of the proof can be restored with the help of Appendix B in reference [16], where the OAM of an sLG beam has been found. Note that when the amplitude parameter is switched off, $\epsilon = 0$, the OAM is described by a simple relation, $\ell_z = \ell \cos 2\beta$, that is inherent in hybrid HLG beams [8], with a control parameter $\pi/4 - \beta$.

An alternative way for the OAM evaluation is when we have a series expansion of the field $F(\mathbf{r})$. Then, both integrals in (9) may be reduced to series or finite sums. In particular, knowing the asLG beam expansion into the HG basis, as in Equation (6), one can find the OAM by utilizing Equation (9) in reference [23]. We can also use the representation of an

HLG mode on the basis of another HLG modes, Equations (21) and (22) in reference [8], with the aim of obtaining the asLG beam expansion into the LG basis:

$$\text{asLG}(\mathbf{r}, \epsilon, \theta, \beta) = \sum_{k=0}^{2n+\ell} b_k(\epsilon, \theta, \beta) \text{LG}_{\min, 2n+\ell-2k}(\mathbf{r}), \tag{12}$$

where $\min = \min(2n + \ell - k, k)$, $\max = \max(2n + \ell - k, k)$ and the coefficients $b_k$ are quite cumbersome (see Appendix B).

The expression obtained enables us to write the OAM in the form

$$\ell_z = \frac{\sum\limits_{k=0}^{2n+\ell} (2n + \ell - 2k)|b_k(\epsilon, \theta, \beta)|^2 \|\text{LG}_{\min, 2n+\ell-2k}(\mathbf{r})\|^2}{\sum\limits_{k=0}^{2n+\ell} |b_k(\epsilon, \theta, \beta)|^2 \|\text{LG}_{\min, 2n+\ell-2k}(\mathbf{r})\|^2}, \tag{13}$$

where $\|\text{LG}_{n,\pm m}(\mathbf{r})\|^2 = \frac{\pi}{2} \cdot \frac{(n+m)!}{2^m n!}$ is the LG beam power.

The unexpected results of computer simulation and experiment presented in Figure 3 surprised us. We found that in the vicinity of $\theta = \pi$, where the OAM for asLG beams turns to zero, $\ell_z(\theta = \pi) = 0$, its sharp bursts and dips (red curves) occur, which are not inherent in astigmatic-free sLG beams (blue curves). In contrast to sLG beams, for which the OAM vanishes at the point $\theta = \pi$, and its maximum does not exceed TC $= \ell$ for all $\theta \in (0, 2\pi)$, the OAM of corresponding asLG beams reaches its maximum in the vicinity of $\theta = \pi$ and the maximum value turns out to be significantly greater than $\ell$. Computer simulations in Figures 3 and 4 show the behavior of the OAM of asLG beams. It is seen that the OAM maximum value depends strongly on the radial number $n$. For example, if $\ell = 1$ and $n \gg 1$, then the value is approximately equal to $(n + 1)/2$. As the radial number $n$ increases, the interval between the OAM bursts and dips on both sides of the point $\theta = \pi$ narrows rapidly. Sharp bursts and dips of the OAM disappear at zero radial number, $n = 0$. When the azimuthal number exceeds the radial number $\ell > n$, the situation changes. As before, with large radial numbers $n \gg 1$, there are splashes and dips of the OAM, but they exchange their places, as can be seen from Figure 4d. However, now their tips can stitch the axis $\ell_z(\theta) = 0$, penetrating deeply into areas with opposite OAM signs. Nevertheless, the TC modulus of the entire sLG beam did not change, since the total number of HG modes remained the same [24].

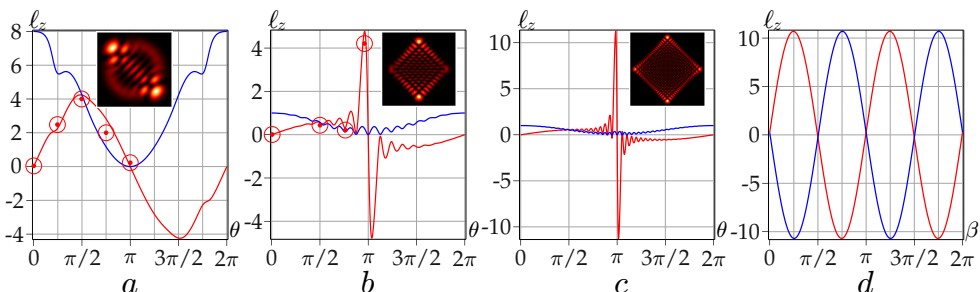

**Figure 3.** The OAM $\ell_z$ of astigmatic-invariant sLG beams and its super bursts in the asLG beams as a function of the phase parameter $\theta$ (**a–c**) and the astigmatic parameter $\beta$ (**d**). In the frame (**d**), $\ell_z(\beta)$ for $\theta = \pi - 0.05$ (red curve) and $\theta = \pi + 0.05$ (blue curve) with $(n, \ell) = (20, 1)$. Other frames depict the curves $\ell_z(\theta)$ for $\beta = 0$ (blue curve) and $\beta = \pi/4$ (red curve) with $(n, \ell) = (0, 8)$ (**a**), $(8, 1)$ (**b**), and $(20, 1)$ (**c**). The callouts demonstrate intensity patterns corresponding to the main maxima of the OAM. Experimental points are indicated by circles.

The astigmatic invariance condition, Condition (7), can be extended to the cases of an amplitude parameter other than unity, $\epsilon \neq 1$. Let us pay attention to the intersection points

of the OAM curves $\ell_z(\theta)$ for astigmatic-free (blue curve) and astigmatic (red curve) beams in Figure 3b–d specified by the condition

$$\Delta\ell_z = \ell_z(\epsilon, \theta, \beta) - \ell_z(\epsilon, \theta, 0) = 0. \tag{14}$$

Excluding the singular point $\theta = \pi$ (where a simple astigmatism vanishes), at the rest points of the blue and red curves, the phase $\theta$ and astigmatic $\beta$ parameters in Equation (14) must satisfy the condition of the astigmatism invariance (14).

To test the assumption (14), intensity patterns (see experimental section) of astigmatism-free sLG and asLG beams (Figure 5a–c) were measured at various amplitude parameters $\epsilon$, and then the correlation degree of these intensity patterns was calculated.

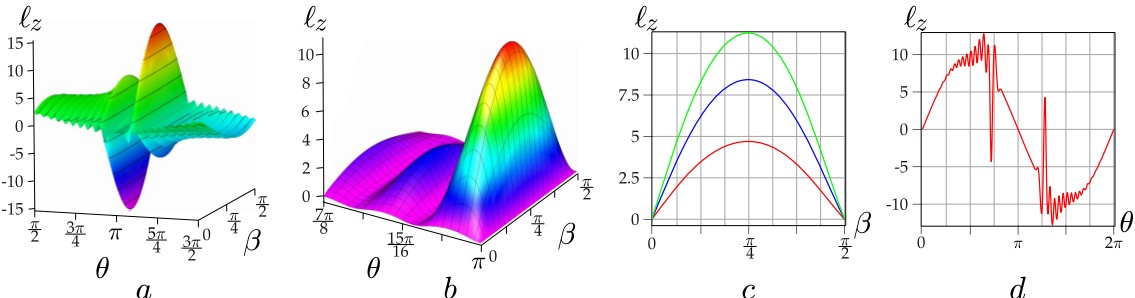

**Figure 4.** (**a**,**b**) 3D intensity distributions of the OAM $\ell_z(1, \theta, \beta)$ for the asLG beam states (**a**) $(n, \ell) = (20, 5)$, (**b**) $(20, 1)$, (**c**) the OAM $\ell_z(\beta)$: red curve—$(n, \ell) = (8, 1)$ for $\theta = \pi - 0.1$, blue curve—$(n, \ell) = (15, 1)$ for $\theta = \pi - 0.075$, green curve—$(n, \ell) = (20, 1)$ for $\theta = \pi - 0.05$ that correspond to max $\ell_z(\beta)$ and (**d**) $(n, \ell) = (15, 20)$, $\beta = \pi/4$.

The comparison shows that more than 10 intensity patterns obtained from Condition (14) for various $\epsilon$ parameters have the correlation degree exceeding 0.93, which convinced us of employing Condition (14).

The curves $\ell_z(\epsilon, \theta, \beta) = 0$ in Figure 5 set values $(\theta, \beta)$ for various amplitude parameters $\epsilon$, and the dependencies $\ell_z(\theta)$ in Figure 5a–c describe the OAM for the corresponding $\epsilon$ parameters. Figure 5 sets the ratio between the phase $\theta$ and astigmatic $\beta$ parameter of the asLG beam for $\epsilon = 1$ in the form of a linear dependence $\theta = 2\beta$ plotted in according to Equation (14). The vertical lines indicate the positions of the OAM zeros. A slight deviation from a unit parameter to $\epsilon = 0.99$ leads to straight line breaks in locations of the former OAM zeros. However, there is no multi-valuedness. The points of discontinuities indicate that in the region of the former OAM zeros, the asLG beams transit into new structurally stable states with a new pair of control parameters $(\theta, \beta)$, but there are no astigmatic-invariant asLG beams at the points of discontinuities.

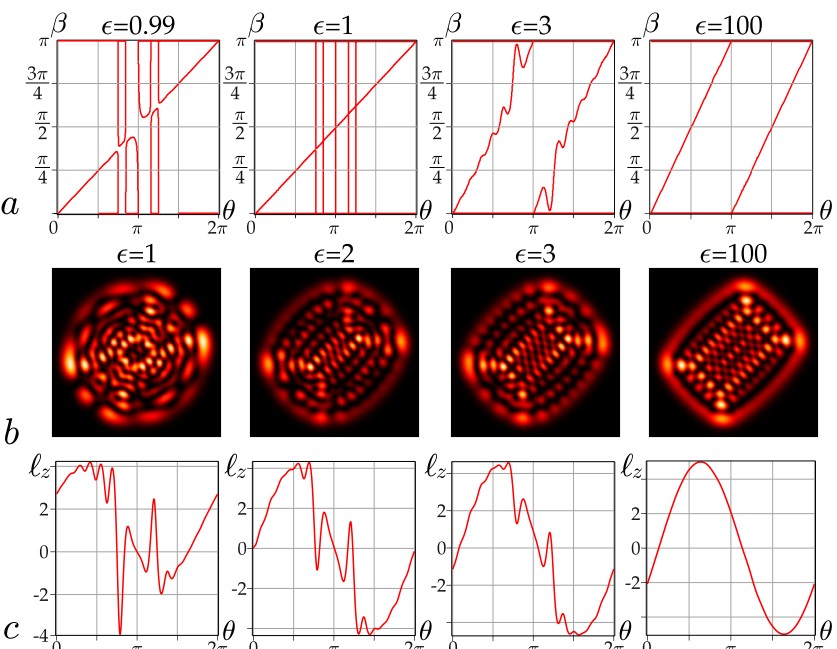

**Figure 5.** (**a**) Invariants of a simple astigmatism at different amplitude parameters $\epsilon$; (**b**) intensity distribution of the asLG beam with $(n, \ell) = (5, 5)$ at different amplitude parameters $\epsilon$ and $(\theta, \beta) = (1, 0.5)$; (**c**) the OAM $\ell_z(\theta)$ of the asLG beam for some values of $\epsilon$ and $\beta = \pi/4$.

As the $\epsilon$ parameter increases, the sharp sections of the curves $\Delta \ell_z(\epsilon, \theta, \beta) = 0$ are smoothed out; the only break point is located at the point $\theta = \pi$, so that at the parameter $\epsilon = 100$, the curves describe the astigmatism of the hybrid HLG beam [17] with the astigmatic-invariant condition $\theta = \beta$, $\theta \in (0, \pi)$. The OAM is described by the equation $\ell_z = \ell \sin \theta$ (see Figure 5b).

## 3. The OAM Sharp Splashes and Dips

It is important to understand the physical origins of the OAM sharp splashes and dips in the vicinity of their zeros in the astigmatic asLG beams, the heights of which are significantly greater than those of the that extremes in the structured sLG beams [16], and their heights can be controlled by the radial number $n$. The answer to this question is hidden in the spectra of LG mods, which control the OAM. The typical development of the spectra shape with a changing the phase parameter $\theta$ is shown in Figure 6. Note that the index $k$ in the mode spectra postponed along the abscissa axis, so that the azimuthal number is calculated as $\mu = 2n + \ell - 2k$ and the radial number as $\nu = \min(2n + \ell - k, k)$. From here it can be seen that the total number of LG modes is the same as the a number of HG modes in Equation (1), and their TC are located in the interval $TC \in (-[2n + \ell], 2n + \ell)$. Figure 6a represents the mode spectra of the astigmatic beam in the state $(n, \ell) = (8, 1)$, where the first spectrum describes the astigmatic-invariant asLG beam, and the remaining spectra describe a sequence of additional OAM maxima and minima before the main OAM maximum (see Figure 3c).

Figure 6b specifies the mode spectra at the OAM zero in the states $(n, \ell) = (3, 1), (8, 1)$ and $(20, 1)$, where $(n, \ell) = (3, 1)$ and $(8, 1)$ are obtained experimentally; the remaining spectra are the result of computer simulations. It immediately catches the eye that in the mode spectra (Figure 6a), both for additional extremes and in an astigmatic-invariant beam, there is a mismatch of mode amplitudes, resulting in a small OAM, no higher than $\ell = 1$. However, in the vicinity of the phase parameter $\theta = \pi$, the LG mode spectra are strictly aligned: to the left of the $\theta = \pi$, most energy is concentrated in the LG mode with the maximum azimuthal number (with $k = 0$) so that $TC = 2n + \ell$ and the radial number $n = 0$, whereas to the right of the $\theta = \pi$, most energy is concentrated in the LG mode with the minimum azimuthal number (with $k = 2n + \ell$) so that $TC = -(2n + \ell)$ and the radial

number $n = 0$. Such matching of mode phases and amplitudes leads to a sharp surge or dip of the OAM.

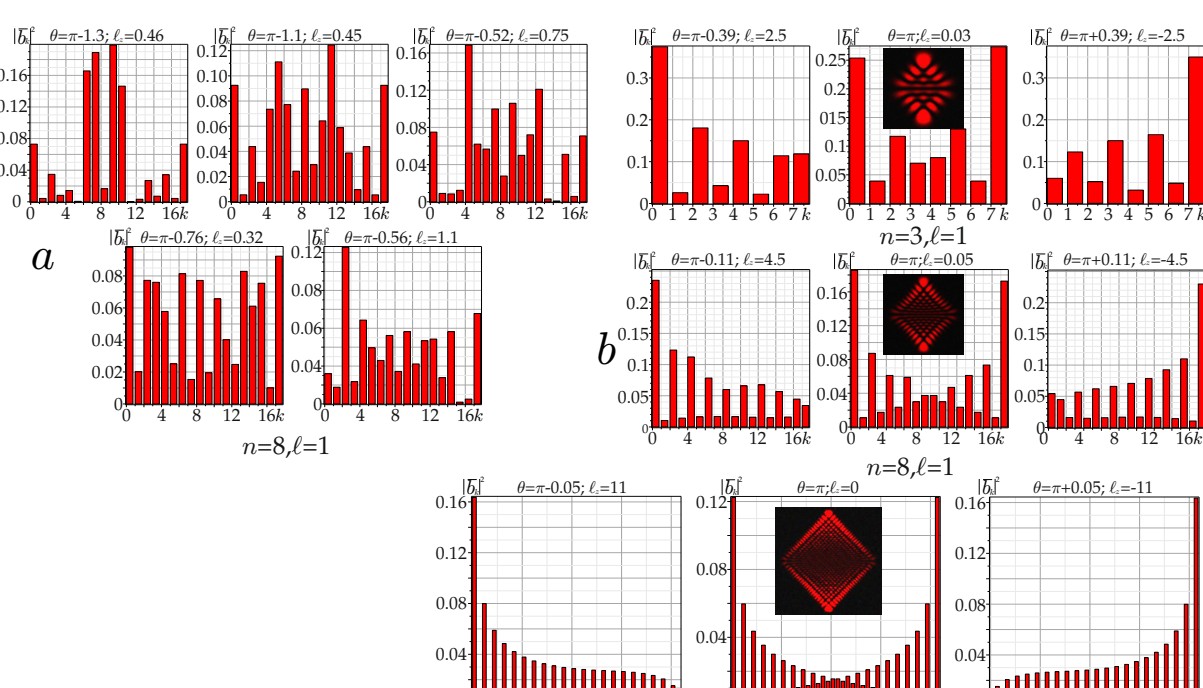

**Figure 6.** The LG mode spectra $b_k(1, \theta, \pi/4)$ of asLG beams for some values of $\theta$: (**a**) LG mode spectra far from main extremes; the first spectrum corresponds to the astigmatic invariant sLG beam with $(n, \ell) = (8, 1)$, and the remaining spectra correspond to a sequence of additional maxima and minima preceding the main maximum. (**b**) The LG mode spectra extremes in the vicinity of zero OAM. The inscriptions in the figures set the phase parameters $\theta$ as the maxima and minima, and the corresponding OAM $\ell_z$. The callouts show intensity patterns for zero OAM. The spectra for $(n, \ell) = (3, 1)$ and $(8, 1)$ are experimental ones.

A slight deviation of the phase parameter $\theta$ from the extreme values leads to the symmetrical restructuring of the LG mode amplitudes with opposite TC, and the OAM turns to zero.

In practice, it is important to know the position of the main maxima of the OAM. Since they are controlled by the radial number $n$ of the asLG beam, it makes sense to simplify cumbersome factors $U, V$ in Equation (11) by replacing them with asymptotical expressions for large radial number, $n \gg 1$. To do this, we employ Equation (8.21.10) from reference [25]: If $\alpha$ and $\beta$ are arbitrary real numbers, then

$$\left(\sin \tfrac{\theta}{2}\right)^{\alpha} \left(\cos \tfrac{\theta}{2}\right)^{\beta} P_n^{(\alpha, \beta)}(\cos \theta) = \frac{\cos\left\{ (2n + \alpha + \beta + 1)\tfrac{\theta}{2} - (2\alpha + 1)\tfrac{\pi}{4} \right\}}{\sqrt{\tfrac{\pi}{2} n \sin \theta}} + \mathcal{O}(n^{-3/2}), \quad (15)$$

where $0 < \theta < \pi$. Using this result, we come to approximate expression for the OAM of the asLG beams, Equation (10), where

$$V = \frac{\cos\left\{ (2n + \ell + 1)\tfrac{\theta}{2} - \tfrac{\pi}{4} \right\}}{\sqrt{\tfrac{\pi}{2} n \sin \theta}}, \qquad U = V \cdot \frac{\cos\left(\tfrac{\theta}{2} - 2\beta\right)}{\cos \tfrac{\theta}{2}}. \quad (16)$$

The appearance of sharp peaks of the OAM in Figure 3 is accompanied by a rather dramatic scenario of interference competition of LG modes in an astigmatic beam. As we

have already noted, sharp splashes and dips of the OAM occur in the vicinity of the phase parameter $\theta = \pi$ where the OAM vanishes, and the heights (depth) of the peaks depend on the radial number $n$: $\max \ell_z \approx (n+1)/2$, where $\ell = 1$ and $n \gg 1$. At first glance, it seems that in order to find the asymptotic value of the OAM, it is sufficient to require $n \to \infty$ in Equation (10) and for parameters $U$ and $V$ in Equation (16). However, the denominators of these parameters include a factor $\sqrt{n} \sin \theta$, whereas the peaks of the OAM appear when $n \to \infty$, $\theta \approx \pi$, so that only with their optimal ratio are there splashes and dips of the OAM at the parameters $\theta^{(\max)}$ and $\theta^{(\min)}$. A rather cumbersome derivation of the optimal phase parameter is

$$\theta_m^{(\max)} \approx \pi - \frac{1}{2n + \ell + \frac{1}{2}}\left(\frac{5\pi}{4} + 2\pi m - \frac{3(2+\sqrt{2})}{\pi(8m+5)}\right) + \mathcal{O}(n^{-2}), \quad m = 0, 1, \ldots , \quad (17)$$

for the maximum OAM carry over to Appendix C. Near this phase parameter $\theta \approx \pi$, a fast ordering of the LG mode spectrum takes place, which causes sharp peaks of the OAM in Figures 3 and 6 (see next Section).

As follows from Equation (17), the positions of the OAM maxima do not depend on the astigmatic parameter $\beta$. However, the higher terms of the asymptotics take this dependence into account. The exact calculation (see Equation (14)) of the max(OAM) positions $\theta_0^{(\max)}$ on the astigmatic parameter $\beta$, specified by the curves in Figure 7b,d, highlight small shifts of the OAM maxima for large radial numbers $n \gg 1$, but they are very difficult to measure experimentally.

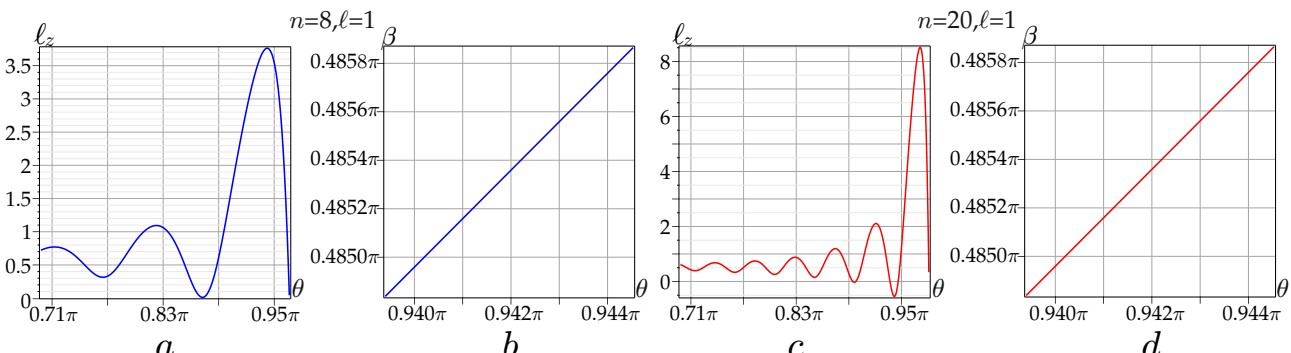

**Figure 7.** Asymptotic plots of the OAM (**a**,**c**) for $(\epsilon, \beta) = (1, \pi/4)$ and positions of main maxima (**b**,**d**) at $\epsilon = 1$ in the asLG beams with the indices $(n, \ell) = (8, 1)$ (**a**,**b**) and $(n, \ell) = (20, 1)$ (**c**,**d**).

## 4. Experiment

The main tasks of our experimental research were, first of all, to confirm shaping astigmatic-invariant beams when Condition (7) is fulfilled, and to reveal sharp bursts of the OAM in the astigmatic structured asLG beams. We used the experimental setup depicted in Figure 8, which was previously described in detail in our article [16], with the only difference being the control of the beam waist radius $w_0$ on the cylindrical lens. We recall (see reference [17,21] and references therein) that with simple astigmatism, when the axis of the cylindrical lens coincides with the axis $y$, a phase factor appears (just behind the lens) in the form $\exp(-ikx^2/2f)$, where $f$ is a focal length of the cylindrical lens (see, e.g., reference [21]).

In our theoretical calculations, the $x, y$ coordinates were normalized by the waist radius $w_0$, i.e., $kx^2/2f = kw_0^2(x/w_0)^2/2f \to (z_0/f)x^2$, where $z_0 = kw_0^2/2$, and we left the designation $x/w_0 \to x$. In our experiment, the Rayleigh length $z_0$ was specified by the waist radius $w_0$ of the Gaussian beam on the entrance pupil of the cylindrical lens. By adjusting the waist radius $w_0$ of the Gaussian beam on the SLM modulator and at the plane of the cylindrical lens by SLF 1 ans SLF2, we achieved the optimal degree of correlation of the astigmatic-invariant asLG beam and the structured LG beam without the astigmatic transform. We believe that for an optimal coincidence of the intensity pattern

of the astigmatic-invariant LG beam and an astigmatic-free LG beam, it is sufficient that their degree of correlation is not lower than 0.93. This choice of the optimal value 0.93 is related to the features of our experimental setup. The fact is that the accuracy of measuring the waist radius $w_0$ of the Gaussian beam is limited due to inaccuracies in the alignment of the optical system. In particular, this affects the measurement of the positions of the main OAM maxima of the astigmatic asLG beams (as we remark in the discussion of Figure 6b), when we cannot distinguish the close positions of the OAM maxima with different radial numbers. The waist radius of the Gaussian beam at the cylindrical lens plane was calculated by measuring the second-order intensity moments $J_{02} = w_x^2$ and $J_{20} = w_y^2$ so that $w_0 = (w_x + w_y)/2$, which was considered in detail in our articles [26,27]. Another important limitation of our measurement process is the resolution of the SLM modulator and the photodetector CMOS.

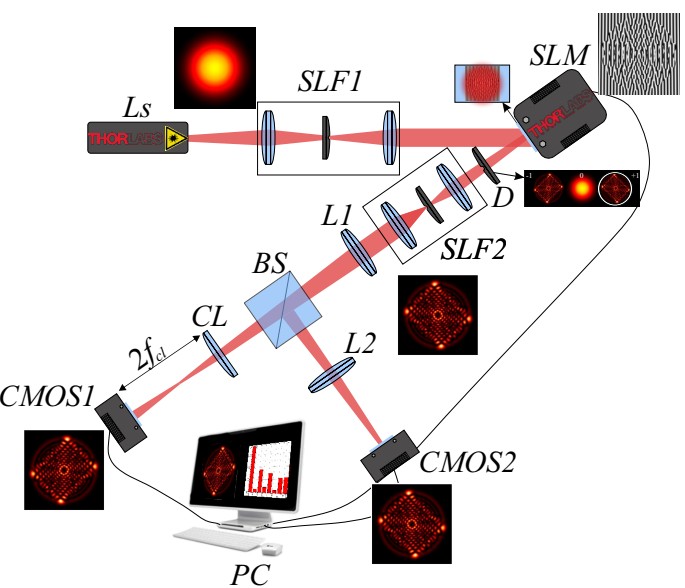

**Figure 8.** Sketch of the experimental setup: Ls—He–Ne laser ($\lambda = 0.633$ μm), SLF1,2—spatial light filter, SLM—spatial light modulator, L1, L2—spherical lenses with $f_{sp} = 25$ cm (focus length), BS—beam splitter, CL—cylindrical lens with $f_{cl} = 15$ cm, CMOS1,2—complementary metal-oxide-semiconductor detector, PC—computer.

In our experiment, we used a reflective SLM with an active liquid crystal to generate sLG beams, which consisted of layers of optically anisotropic molecules. Due to some inevitable inhomogeneity within the liquid crystal, scattering occurred, which reduced the overall reflectivity [28]. The EXULUS-4K1 SLM used by us has a fairly high fill factor (>93%) of pixels (pixel size 3.74 μm) in a liquid crystal cell, and a high level of average reflectivity of the SLM (75%) is a prerequisite for high diffraction efficiency, which makes it possible to reduce the effect of aliasing between adjacent liquid crystal display (LCD) pixels. An important factor in our experiment during the generation of structured beams was the angle of incidence of the laser beam on the LCD screen of the modulator. It is known [28] that the angle of the incident light beam on the SLM also affects the phase delay of the SLM, changing the contrast of the SLM. In our case, the angle of incidence of the laser beam did not exceed 30 degrees from the normal, in which case the change in the diffraction efficiency is less than 5%. The resolution of our EXULUS-4K1 SLM ($3840 \times 2160$, UHD) makes it possible to reproduce and detect beams containing more than 100 LG modes. The beam splitter (Thorlabs BS007 50:50) divided the sLG beam into two optical arms. In the first (straight) arm, the LG mode spectra at the plane of the double focus $2f$ of the cylindrical lens CL were measured in accordance with the intensity moments technique considered in the article [27], which allowed us to take into account the signs of the TC optical vortices in the astigmatic asLG beam, and in the second arm the intensity

distribution in the astigmatic-free sLG beam was analyzed. The measured LG mode spectra near the main maxima, minima and zeros of the OAM are depicted in Figure 6b for the beam states $(n, \ell) = (3, 1)$ and $(8, 1)$. At the OAM maximum, most energy is concentrated in the main LG modes with $(n, \ell) = (0, 3)$ and $(0, 8)$, respectively; the energy of the rest modes decreases rapidly throughout the range from TC = 2 and TC = 8 to TC = $-2$ and TC = $-8$. The reverse situation can be observed for the OAM minimum. Most energy is concentrated in modes with $(n, \ell) = (0, -3)$ and $(0, -8)$, respectively. In the OAM zero, the energy is distributed approximately equally between all LG modes with positive and negative TC. These mode spectra allowed us to calculate the OAM of the beams in accordance with Equation (13). The locations of the experimental points in Figure 3 indicate good agreement between theory and experiment.

As already noted, to test Condition (7), under which astigmatic-invariant beams were formed, the intensity patterns in the first and second arms of the experimental setup were analyzed, and then the patterns were scaled and compared with each other. Typical intensity distributions of the asLG and sLG beams are shown in Figure 2. The degree of correlation of these intensity patterns was not lower than 0.93, which indicates that Condition (7) was fulfilled.

## 5. Discussion and Conclusions

When a structured sLG beam passes through an optical astigmatic element, for example, through a cylindrical lens or a spherical lens whose axis does not coincide with the axis of the optical system, its internal structure radically changes. Such an astigmatic transform increases the number of degrees of freedom of the structured beams and expands the area of their technical applicability. However, in some cases it is necessary that the astigmatic element does not change the light beam's structure. However, there are some key points to be emphasized. The first key point relates to the astigmatic invariance condition in Equation (7) for $\epsilon = 1$ and Equation (14) for arbitrary $\epsilon \neq 1$. This means that it is always possible to choose the phase parameter $\theta$ of a structured LG beam in such a way that its structure will be reproduced with accuracy up to that of linear scale transforms in an astigmatic medium. Therefore, at first glance it seems that this also applies to a separate LG mode, since it can be represented as a superposition of the HG modes in Equation (1). However, this is not the case. The fact is that the astigmatic invariance of the structured beam means a certain rearrangement of the HG modes in Equation (8) with strictly matched phases that cannot be done in Equation (1). Of course, it is possible to require that the OAM in the astigmatic-free LG mode for $\epsilon = 0$ coincides with the OAM of the structured asLG beam with $\epsilon \neq 0$ and obtain a finite set of phase parameters $\theta$. However, Equation (14) will not be fulfilled, since the first and second terms in Equation (14) have different amplitude parameters $\epsilon$, and consequently, it is impossible to obtain the condition of astigmatic invariance of a separate LG mode.

Thus, we revealed theoretically and experimentally confirmed that the conditions of the astigmatic invariance for arbitrary radial $n$ and azimuthal $\ell$ numbers are that the phase parameter $\theta$ of the sLG beam is equal to the arctangent of the ratio of the Rayleigh length $z_0$ and the focal length $f$ of the cylindrical lens for the unit amplitude parameter $\epsilon = 1$. For the remaining values of the amplitude parameter $\epsilon$, the astigmatic invariance condition is specified by the equality of the OAM of an sLG and astigmatic-invariant asLG beams.

The second key point concerns sharp splashes in OAM near a special point of the phase parameter $\theta = \pi$, where astigmatism disappears. The larger the radial number $n$, the higher the height of the OAM bursts, but at the same time, the width $\Delta\theta$ of the OAM splash contour sharply narrows. At the same time, the position of the OAM maximum does not depend on the astigmatism parameter $\beta$ in the first order of smallness in the radial number $n$, and only in the second order of smallness is there dependence on the $\beta$. We have linked such an acute OAM splash with a sharp restructuring of the LG mode spectrum in Figure 6b, that is, with the OAM of the eigen LG modes of the structured beam. However, in a number of articles (see, e.g., reference [29]), it was shown that vortex-free

beams acquire the OAM in astigmatic media due to a sharp elongation of the beam intensity distribution in one direction. However, a comparison of the of the intensity patterns at the OAM maxima in the callouts of Figure 3b–d with the intensity patterns at zero OAM shows that in Figure 6b there is no sharp changes in the symmetry of astigmatic beams. Therefore, we concluded that the main contribution to the OAM splashes is the ordered spectrum of optical vortices in Figure 6b. Nevertheless, the method of calculating the exact contribution to sharp OAM bursts due to the displacement of the center of gravity, astigmatism or optical vortices was considered in reference [30] based on the approach of second-order intensity moments. Such a cumbersome calculation goes far beyond the scope of this article but is of interest for further research.

Thus, we also found sharp bursts and dips of the OAM in astigmatic asLG beams in the area where OAM turns to zero, and the heights and depths of these bursts and dips significantly exceed the OAM maxima and minima in conventional structured sLG beams. It has been shown that OAM bursts and dips are caused by strong restructuring of the mode spectrum in the form of rigid ordering of the LG modes. The theoretical calculation, computer simulation and experiment agreed well with each other.

A few words should be said about the practicality of employing the OAM sharp bursts in the asLG beam. As is well known, the spin moment of the beam transmits a rotational movement to the particles around their axes, regardless of where the particle is in the beam plane. The OAM behaves quite differently. Rotation around its own axis is transmitted to particles located on the beam axis only. If the particle is displaced relative to the beam axis, it receives a torque around the beam axis. This property can be used to sort particles by their masses with gradual displacement of the particles relative to the beam axis. The phase parameter $\theta$ of the beam can be changed smoothly in the vicinity of the OAM burst using the SLM device, thereby gradually increasing the particle's outgoing momentum until is tears off the circular trajectory. Such a system can act similar to a mass spectrograph.

**Author Contributions:** Conceptualization, A.V.; software, M.B.; validation, E.A. and A.V.; formal analysis, M.B.; investigation, A.V. and Y.A.; writing—original draft preparation, A.V. and E.A.; writing—review and editing, A.V.; visualization, M.B. and Y.A. All authors have read and agreed to the published version of the manuscript.

**Funding:** This work was supported by the Russian Foundation for Basic Research (grants 20-37-90066, 20-37-90068 and 19-29-01233) and the Ministry Council of Republic of Crimea.

**Institutional Review Board Statement:** Not applicable.

**Informed Consent Statement:** Not applicable.

**Data Availability Statement:** The data presented in this study are available on request from the corresponding author.

**Conflicts of Interest:** The authors declare no conflict of interest.

## Appendix A. Astigmatic Transform of sLG Beams

Here we obtain the expression of the astigmatically transformed sLG beam in terms of HLG modes as it is written in Equation (A11). We start with a particular case of the general astigmatic transformation of an HLG mode, considered in reference [17], when the defocusing component is absent:

$$
\begin{aligned}
&\frac{1}{2\pi} \int_{\mathbb{R}^2} \exp\big\{ -i\langle \mathbf{r}, \boldsymbol{\rho} \rangle + ib\psi(\boldsymbol{\rho}, \phi) \big\} \mathrm{HLG}_{n,m}(\boldsymbol{\rho} \mid \alpha) \mathrm{d}^2\boldsymbol{\rho} \\
&= \frac{(-i)^{n+m}}{2\sqrt{1+b^2}} \exp\left\{ -\frac{ib\psi(\mathbf{r},\phi)}{4(1+b^2)} + i(n-m)\varphi \right\} \mathrm{HLG}_{n,m}\left( \frac{\mathbf{R}_{-\gamma-\phi}\mathbf{r}}{2\sqrt{1+b^2}} \,\Big|\, \vartheta \right).
\end{aligned}
\tag{A1}
$$

The parameters $\gamma, \vartheta, \varphi$ are solutions of the following system of equations:

$$\sin 2\vartheta = \sin 2\alpha \cos 2\beta + \cos 2\alpha \sin 2\phi \sin 2\beta,$$

$$e^{2i\gamma} \cos 2\vartheta = \cos 2\alpha \cos 2\phi + i(\sin 2\alpha \sin 2\beta - \cos 2\alpha \sin 2\phi \cos 2\beta),$$

$$e^{i\varphi \pm i\vartheta} = e^{i\beta}(\cos\gamma \mp \sin\gamma)(\cos\alpha\cos\phi + i\sin\alpha\sin\phi)$$
$$- e^{-i\beta}(\sin\gamma \pm \cos\gamma)(\cos\alpha\sin\phi - i\sin\alpha\cos\phi), \qquad (A2)$$

where $\beta = \arctan b$. It should be noted that only one solution of the system (A2) is needed because any other leads to the same right-hand side of Equation (A1).

Since we are interesting in the astigmatically transformed sLG beam with the astigmatic influence function $\psi(\boldsymbol{\rho}, 0)$, it is necessary to consider the formulae (A1) and (A2) for the modes $\mathrm{HLG}_{n+\ell,n}(\boldsymbol{\rho} \,|\, \pi/4)$ and $\mathrm{HLG}_{n,n+\ell}(\mathbf{R}_{-\pi/4}\boldsymbol{\rho} \,|\, \Theta)$.

For the first HLG mode, $\alpha = \pi/4$ and $\phi = 0$. Then, the system (A2) has the solution $(\gamma, \vartheta, \varphi) = (\pi/4, \pi/4 - \beta, \pi/4)$, and we have

$$\frac{1}{2\pi} \int_{\mathbb{R}^2} \exp\{-i\langle \mathbf{r}, \boldsymbol{\rho}\rangle + ib\psi(\boldsymbol{\rho}, 0)\} \mathrm{HLG}_{n+\ell,n}(\boldsymbol{\rho} \,|\, \pi/4) \mathrm{d}^2\boldsymbol{\rho}$$
$$= \frac{(-i)^{2n+\ell}}{2\sqrt{1+b^2}} \exp\left\{-\frac{ib\psi(\mathbf{r}, 0)}{4(1+b^2)} + \frac{i\ell\pi}{4}\right\} \mathrm{HLG}_{n+\ell,n}\left(\frac{\mathbf{R}_{-\pi/4}\mathbf{r}}{2\sqrt{1+b^2}} \,\Big|\, \frac{\pi}{4} - \beta\right). \qquad (A3)$$

For the second HLG mode, we first rotate the integration variables, $\boldsymbol{\rho} \to \mathbf{R}_{\pi/4}\boldsymbol{\rho}$, and then use Equations (A1) and (A2) with $\alpha = \Theta$ and $\phi = -\pi/4$:

$$\frac{1}{2\pi} \int_{\mathbb{R}^2} \exp\{-i\langle \mathbf{r}, \boldsymbol{\rho}\rangle + ib\psi(\boldsymbol{\rho}, 0)\} \mathrm{HLG}_{n,n+\ell}(\mathbf{R}_{-\pi/4}\boldsymbol{\rho} \,|\, \Theta) \mathrm{d}^2\boldsymbol{\rho}$$
$$= \frac{1}{2\pi} \int_{\mathbb{R}^2} \exp\{-i\langle \mathbf{R}_{-\pi/4}\mathbf{r}, \boldsymbol{\rho}\rangle + ib\psi(\boldsymbol{\rho}, -\pi/4)\} \mathrm{HLG}_{n,n+\ell}(\boldsymbol{\rho} \,|\, \Theta) \mathrm{d}^2\boldsymbol{\rho}$$
$$= \frac{(-i)^{2n+\ell}}{2\sqrt{1+b^2}} \exp\left\{-\frac{ib\psi(\mathbf{R}_{-\pi/4}\mathbf{r}, -\pi/4)}{4(1+b^2)}\right\} \mathrm{HLG}_{n,n+\ell}\left(\frac{\mathbf{R}_{-\pi/4}\mathbf{r}}{2\sqrt{1+b^2}} \,\Big|\, \Theta - \beta\right). \qquad (A4)$$

Here, the system (A2) has the solution $(\gamma, \vartheta, \varphi) = (\pi/4, \Theta - \beta, 0)$.

Since $\psi(\mathbf{R}_{-\pi/4}\mathbf{r}, -\pi/4) = \psi(\mathbf{r}, 0)$, the astigmatic phase factors of right-hand parts of Equations (A3) and (A4) are the same. By combining the integrands in left-hand parts of both equations into the sLG beam, one can find

$$\frac{1}{2\pi} \int_{\mathbb{R}^2} \exp\{-i\langle \mathbf{r}, \boldsymbol{\rho}\rangle + ib\psi(\boldsymbol{\rho}, 0)\} \mathrm{sLG}_{n,\ell}(\boldsymbol{\rho}, \epsilon, \theta) \mathrm{d}^2\boldsymbol{\rho}$$
$$= \frac{(-i)^{2n+\ell}}{2\sqrt{1+b^2}} \exp\left\{-\frac{ib\psi(\mathbf{r}, 0)}{4(1+b^2)}\right\} \cdot \frac{(-1)^n}{2^{n+\ell}n!}\left\{e^{i\ell\pi/4}\mathrm{HLG}_{n+\ell,n}\left(\frac{\mathbf{R}_{-\pi/4}\mathbf{r}}{2\sqrt{1+b^2}} \,\Big|\, \frac{\pi}{4} - \beta\right)\right.$$
$$\left. + \epsilon(-i)^{n+\ell}e^{i(2n+\ell)\Theta}\mathrm{HLG}_{n,n+\ell}\left(\frac{\mathbf{R}_{-\pi/4}\mathbf{r}}{2\sqrt{1+b^2}} \,\Big|\, \Theta - \beta\right)\right\}, \qquad (A5)$$

which coincides with Equation (5).

## Appendix B. Expansion of an asLG Beam into the Series of LG Modes

Here we obtain the coefficients $b_k$ used in Equation (12). We start with Equation (21) in [8], which provides the expansion of an HLG mode with some parameter into a series of HLG modes with another parameter:

$$\mathrm{HLG}_{n,m}(\mathbf{R}_\gamma \mathbf{r}\,|\,\theta) = \sum_{k=0}^{n+m} \Lambda_k^{(n,m)} \mathrm{HLG}_{n+m-k,k}(\mathbf{r}\,|\,\alpha), \tag{A6}$$

$$\Lambda_k^{(n,m)} = (-1)^k \{\cos\gamma\cos(\theta-\alpha) + \mathrm{i}\sin\gamma\sin(\theta+\alpha)\}^{n-k} \times$$
$$\times \{\cos\gamma\sin(\theta-\alpha) - \mathrm{i}\sin\gamma\cos(\theta+\alpha)\}^{m-k} \times$$
$$\times P_k^{(n-k,m-k)}\big(\sin^2\gamma\cos 2(\theta+\alpha) - \cos^2\gamma\cos 2(\theta-\alpha)\big).$$

Substituting the values $\gamma = -\pi/4$ and $\alpha = \pi/4$ into Equation (A6) yields

$$\Lambda_k^{(n,m)} = (-1)^m e^{-\pi\mathrm{i}(n+m-2k)/4}\sin^{n-k}(\theta+\pi/4)\cos^{m-k}(\theta+\pi/4)P_k^{(n-k,m-k)}(-\sin 2\theta).$$
$$= (-1)^m e^{-\pi\mathrm{i}(n+m-2k)/4}c_k^{(n,m)}\Big(\frac{\pi}{4}-\theta\Big), \tag{A7}$$

where

$$c_k^{(n,m)}(\alpha) = \cos^{n-k}\alpha\,\sin^{m-k}\alpha\,P_k^{(n-k,m-k)}(-\cos 2\alpha)$$
$$= \sum_{j=\max(0,k-m)}^{\min(k,n)} (-1)^{k-j}\binom{n}{j}\binom{m}{k-j}\cos^{n+k-2j}\alpha\,\sin^{m-k+2j}\alpha \tag{A8}$$

are auxiliary coefficients used for brevity.

Then HLG components of the asLG beam, Equation (5), have the following expansions in LG modes:

$$\mathrm{HLG}_{n+\ell,n}\Big(\mathbf{R}_{-\pi/4}\mathbf{r}\,\Big|\,\frac{\pi}{4}-\beta\Big) = (-1)^n e^{-\pi\mathrm{i}(2n+\ell)/4}$$
$$\times \sum_{k=0}^{2n+\ell} \mathrm{i}^k c_k^{(n+\ell,n)}(\beta)\mathrm{HLG}_{2n+\ell-k,k}(\mathbf{r}\,|\,\pi/4),$$
$$\mathrm{HLG}_{n,n+\ell}\big(\mathbf{R}_{-\pi/4}\mathbf{r}\,|\,\Theta-\beta\big) = (-1)^{n+\ell} e^{-\pi\mathrm{i}(2n+\ell)/4}$$
$$\times \sum_{k=0}^{2n+\ell} \mathrm{i}^k c_k^{(n,n+\ell)}\Big(\frac{\pi-\theta}{2}+\beta\Big)\mathrm{HLG}_{2n+\ell-k,k}(\mathbf{r}\,|\,\pi/4), \tag{A9}$$

and the final result is Equation (12) with the coefficients

$$b_k = (-1)^{\min}2^{\max}(\min)!\cdot\frac{\mathrm{i}^{k-n}}{2^{n+\ell}n!}\left\{c_k^{(n+\ell,n)}(\beta) + \epsilon(-1)^n e^{\mathrm{i}(2n+\ell)\theta/2}c_k^{(n,n+\ell)}\Big(\frac{\pi-\theta}{2}+\beta\Big)\right\}. \tag{A10}$$

In the last step, we use the relation between HLG and LG modes [17]:

$$\mathrm{HLG}_{n,m}(\mathbf{r}\,|\,\pi/4) = (-1)^{\min}2^{\max}(\min)!\cdot\mathrm{LG}_{\min,n-m}(\mathbf{r}) \tag{A11}$$

with $\min = \min(n,m)$ and $\max = \max(n,m)$.

### Appendix C. Asymptotic Expressions of Maximum and Minimum Points for the OAM of an asLG Beam

Here we find maximum and minimum points of the OAM, Equation (10), as a function of $\theta$, assuming $\epsilon = 1$ and $n \to \infty$. We restrict ourselves to the case in which $\ell$ is odd, $0 < \theta < \pi$ and $0 < \beta < \pi/2$ for simplicity. The most interesting behavior of the OAM is near $\theta = \pi$, i.e., when $n\sin\theta \sim \infty \cdot 0$. Let us denote $N = 2n + \ell$. By substituting

Equation (16) into (10) and changing the variable $\theta$ to $\vartheta$ by $\theta = \pi - 2\vartheta/(2N+1)$, where $\vartheta > 0$, we have

$$\ell_z\left[\text{asLG}_{n,\ell}(\mathbf{r},\epsilon,\theta,\beta)\right] = \ell \sin\left(2\beta + \tfrac{\vartheta}{2N+1}\right) \cdot \frac{Q + \sin\tfrac{\vartheta}{2N+1}}{1 + Q\sin\tfrac{\vartheta}{2N+1}}, \tag{A12}$$

where $Q$ is a cumbersome fraction:

$$Q = \frac{\cos\left(\tfrac{\pi}{4} - \tfrac{\vartheta}{2N+1}\right) - \cos\left(\tfrac{\pi}{4} - \vartheta\right)}{\sin\tfrac{\vartheta}{2N+1}\sqrt{\pi(N-\ell)\sin\tfrac{2\vartheta}{2N+1}}}. \tag{A13}$$

Since $N$ is a large parameter, both expressions can be simplified with the help of well-known relations $\sin\delta \sim \delta$, $\cos\delta \sim 1$ as $\delta \to 0$. Then,

$$Q \sim \frac{1 - \cos\vartheta - \sin\vartheta}{\tfrac{\vartheta}{2N+1}\sqrt{2\pi\vartheta}},$$

$$\ell_z \sim \ell\sin 2\beta \cdot \frac{Q}{1 + Q\tfrac{\vartheta}{2N+1}} \sim (2N+1)\ell\sin 2\beta \cdot \frac{1 - \cos\vartheta - \sin\vartheta}{\vartheta(\sqrt{2\pi\vartheta} + 1 - \cos\vartheta - \sin\vartheta)}. \tag{A14}$$

We denote the latter fraction by $S(\vartheta)$. Thus, the problem is reduced to getting the solutions of the equation $S'(\vartheta) = 0$.

After deriving the function $S(\vartheta)$, one can write the equation as follows:

$$\frac{(1 - \cos\vartheta - \sin\vartheta)^2}{\vartheta\sqrt{2\pi\vartheta}} + \frac{3}{2}\cdot\frac{1 - \cos\vartheta - \sin\vartheta}{\vartheta} + (\cos\vartheta - \sin\vartheta) = 0, \tag{A15}$$

where all three terms are continuous functions. Numerically, the first roots are $\vartheta_1 = 2.956$, $\vartheta_2 = 7.009$, $\vartheta_3 = 9.904$ and $\vartheta_4 = 13.320$.

For analytic evaluation of the roots of Equation (A15), we use the following arguments. First, when $\vartheta$ is small, all three terms in Equation (A15) are important. However, with growing $\vartheta$, the influence of the first two terms become more and more insignificant due to the denominators presence. Moreover, neglecting the first term, we obtain 7.006 for the value of the second root. Thus, removing the first term from Equation (A15), one can reduce the equation to the following one:

$$\sin\left(\vartheta - \frac{\pi}{4} + \arctan\frac{3}{2\vartheta}\right) = \frac{3}{\sqrt{2(4\vartheta^2 + 9)}}, \tag{A16}$$

which has two series of solutions. The first one is

$$\vartheta - \frac{\pi}{4} + \frac{3}{2\vartheta} = \frac{3}{2\vartheta\sqrt{2}} + 2\pi m \quad\Rightarrow\quad \vartheta + \frac{3(\sqrt{2} - 1)}{2\vartheta\sqrt{2}} = \frac{\pi}{4} + 2\pi m \quad\Rightarrow$$

$$\vartheta_m \approx \frac{\pi}{4} + 2\pi m - \frac{3(2 - \sqrt{2})}{\pi(8m + 1)}, \quad\Rightarrow\quad \vartheta_1 = 7.006, \quad \vartheta_2 = 13.319, \tag{A17}$$

and the second is

$$\frac{5\pi}{4} - \vartheta - \frac{3}{2\vartheta} = \frac{3}{2\vartheta\sqrt{2}} - 2\pi m \quad\Rightarrow\quad \vartheta + \frac{3(\sqrt{2} + 1)}{2\vartheta\sqrt{2}} = \frac{5\pi}{4} + 2\pi m \quad\Rightarrow$$

$$\vartheta_m \approx \frac{5\pi}{4} + 2\pi m - \frac{3(2 + \sqrt{2})}{\pi(8m + 5)}, \quad\Rightarrow\quad \vartheta_0 = 3.275, \quad \vartheta_1 = 9.959. \tag{A18}$$

Here we replace all $\arcsin\delta$ and $\arctan\delta$ values by their arguments $\delta$, assuming that $\delta$ is small. (A numerical check verified this step, except the smallest root, $\vartheta_0$.) Of course, when $m$ is large, both series merge into the one, $\pi/4 + \pi m$. A routine procedure shows that Equa-

tions (A17) and (A18) describe minimum and maximum points of OAM, correspondingly. Returning to $\theta$ we obtain the final expression, Equation (17).

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
