# Peer review of "Astigmatic-Invariant Structured Singular Beams"

_photonics, doi:10.3390/photonics9110842_

Round 1
Reviewer 1 Report
Please see attached for my detailed review comments. Good luck!

Reviewer 2 Report
See attached file

Reviewer 3 Report
The authors studied the properties of the astigmatic-invariant structured singular beams. The results are interesting. This manuscript can be published after following revisions.
1. On line 8, what is the meaning of the asLG beams in the abstract?
2. On line 12, the word inroduction should be replaced by introduction.
3. On line 22, the authors said “The simplest way to transform a standard LG beam into a structured LG …”, the authors should explain the physical meanings of the standard LG beam and structured LG.
4. On line 27-28, the authors said “we employ a two-parameter perturbation function…” the authors should give the physical meanings of the perturbation function.
5. On line 91, what is the meaning of the TC beams.
6. In Eq.5 and Eq.9, what is the meaning of the parameter R2.
7. On line 81, the authors said “…, the superposition of a large number of HG modes leads to rather radical transformations of the intensity pattern, ...” the authors can explain this phenomenon.
8. On line 134-135, the authors said “… for astigmatic-free (blue curve) and astigmatic (red curve) beams in Fig.4b-d specified by the condition.” The Fig.4b-d should be replaced by Fig.3b-d.
9. On line 149, the authors said “… to ε=1.01 leads to straight line breaks in locations of the former OAM zeros.” No figure with ε=1.01 can be found in Fig. 5. The authors should check it carefully.
10. On line 174, the authors said “…, while the spectra(a-f) are obtained experimentally, ….” We do not find spectra(a-f) in Fig. 6b. The authors should check it carefully.
